# Transitional Care for Spinal Cord Injuries in Hong Kong SAR, China: A Narrative Review of the Local Experience

**DOI:** 10.3390/healthcare12232388

**Published:** 2024-11-28

**Authors:** Chor-Yin Lam, Ivan Yuen-Wang Su, Joyce Yuk-Mui Law

**Affiliations:** 1Department of Orthopaedics and Traumatology, School of Clinical Medicine, Li Ka Shing Faculty of Medicine, The University of Hong Kong, Hong Kong SAR, China; 2SAHK, Hong Kong SAR, China; ivan_syw@sahk1963.org.hk (I.Y.-W.S.); joyce_lym@sahk1963.org.hk (J.Y.-M.L.)

**Keywords:** spinal cord injury, rehabilitation, transition in care, healthcare delivery, model of care

## Abstract

**Background:** Spinal cord injuries (SCI) are devastating conditions which often cause multiple permanent physical impairments and psychosocial complications. Discharge from hospital is often delayed and precious health resources are consumed. In Hong Kong SAR, China, the government welfare system and the public hospital system have worked together to address these problems through partnership with non-governmental organizations. An SCI transitional care facility (the Jockey Club New Page Inn, JCNPI) run by a non-governmental organization (SAHK), was inaugurated in 2008. **Objectives:** Review the local experience of the implementation of SCI transitional care in Hong Kong SAR, China. **Methods:** A narrative review of the service model, facilitators and barriers, and future development. Service output and outcomes were evaluated with quantitative and qualitative means. **Results:** The SCI transitional care in Hong Kong provides person-centred transitional care and support, including a time-limited residential rehabilitation, a post-discharge community day rehabilitation programme, and a residential respite care. The current intervention strategy is based on the WHO’s International Classification of Functioning, Disability, and Health (ICF). In the past 16 years, a total of 226 clients were discharged from the residential rehabilitation service. A total of 223 (98.6%) clients have successfully returned to community living. Positive feedback was received from the service users. **Conclusions**: The SCI transitional care has transformed care for SCI patients from the previous biomedical-oriented, hospital-based rehabilitation into a journey with an empowering and participatory approach addressing their biopsychosocial needs. The model has proven to be a key player in the continuum of care and sustainable community reintegration of individuals with SCI.

## 1. Background

Hong Kong is a Special Administrative Region (SAR) of China. Its 7.5 million residents live in a land area of 1105 km^2^. According to unpublished data from the Clinical Data Analysis and Reporting System of the Hospital Authority, which is responsible for the administration of all public hospitals in Hong Kong, the annual incidence of spinal cord injury (SCI) in Hong Kong was approximately 28 cases per million during the 10 years from 2008 to 2017. Occurrence in males and females was at an approximately 2 to 1 ratio. The mean age of SCI was 60.1 years in males and 65.4 years in females [1]. There are about 210 new cases of SCI per year. Most of these individuals with acute SCI will be managed in the public hospital system and receive inpatient rehabilitation in one of the three designated SCI units in Hong Kong. Most of these SCI cases need home modifications and purchase of assistive devices before they can be discharged and return to community living. Since most of the local population live in relatively small apartment units and the median per capita floor area of accommodation is only 16.0 square meters [2], they often need to be relocated to wheelchair-accessible private or government housing units. This has frequently led to prolonged hospital stays before 2008. Not only are precious healthcare resources consumed, but such stays are also detrimental to the morale of individuals with SCI and their meaningful participation in daily life occupations. Other potential barriers for early return to the community are caused by social factors like financial difficulties, lack of confidence of caregivers, and unavailability of family support.

In order to solve the problems faced by the public hospital system and the individuals with SCI, stakeholders including Social Welfare Department, Hospital Authority, and non-governmental organizations (NGO) came together to search for a new model of service which would provide a continuum of care on the SCI journey towards sustainable community reintegration. The result was the first SCI transitional facility in Southeast Asia which was inaugurated in 2008. The SCI transitional facility was named the Jockey Club New Page Inn (JCNPI) because the initial funding was provided by the Hong Kong Jockey Club Charities Trust, and its aim was to empower people with SCI to start a new page in their lives after devastating injuries. The Centre has been operated by SAHK since its opening. SAHK was found in 1963 as “The Hong Kong Association for the Spastic Children”. Over the years, it has developed into one of the largest NGOs in Hong Kong and changed its name to “The Spastics Association of Hong Kong”, providing various community-based adult and paediatric rehabilitation services across the whole lifespan of clients. In 2008, the Association officially changed its name to “SAHK”. The name has been chosen because it reflects the expansion of service beyond the paediatric group, avoids stigmatization of the clients and, at the same time, retains its identity and long history of service in Hong Kong.

## 2. Objectives

Academic literature focusing on model of care, service outputs and outcomes, facilitators of and barriers to service delivery of SCI transitional care in Asia is scarce. In this article, we intend to review the local experience of the implementation of SCI transitional care in Hong Kong SAR, China. 

## 3. Methods

We performed a narrative review of the service model, facilitators and barriers, and future development of transitional care in Hong Kong SAR, China. Service output and outcomes were evaluated with quantitative and qualitative methods. Information on the transitional care service was obtained with approval from the service provider (SAHK). All three authors have been significantly involved in advisory, supervisory and executive roles in local SCI transitional care. CYL is an honorary medical advisor of SAHK. IWYS is the head of professional and programme development of SAHK. JYML is a senior manager of SAHK and former superintendent of JCNPI.

## 4. Results

### 4.1. Service Model

JCNPI was initially introduced as a 5-year pioneer project. It is located at a public housing estate with accessible design in Ma On Shan of Hong Kong SAR, China. There is a shopping mall, a wet market, and a supermarket as well as banking and recreational facilities within the housing estate. All daily necessities and community services are in close proximity. The surroundings also serve as a convenient environment for community living skill training. Adaptive sports and recreation programmes can be organized in outdoor spaces including parks, a basketball court, and a waterfront promenade. A mass transit station and a bus stop with routes served by wheelchair-accessible buses are located within a distance of 300 m, providing the clients and their caregivers with easy access to accessible public transport (Figure 1). 

JCNPI offers three core programmes for bridging the needs of individuals with SCI who have decided to return to community living. Priority is given to cervical and high thoracic SCI. The first programme is a transitional residential service providing life-oriented rehabilitation to enable newly injured individuals to regain a sense of control over their daily lives and habituate themselves to an active post-injury lifestyle before they resume community living. Secondly, we provide a post-discharge community day rehabilitation programme for up to 12 months to reinforce their competency in adaptive living skills and cope with any new problems they have encountered in community living. Refresher training is provided to caregivers when necessary to maintain their knowledge and skills for taking care of the clients. Finally, a residential respite care is available for a maximum of 30 days per year to allow the caregiver to take a break or when the caregiver is unavailable. We believe that these three core programmes spanning the continuum of the rehabilitation journey are essential for sustainable community reintegration following SCI.

### 4.2. Transitional Residential Service

The first service used by our clients is usually the time-defined residential rehabilitation for an expected period of up to 12 months. Stable SCIs who have the intention to return to community living are referred to us by the three designated SCI units of the public hospital system. A pre-admission assessment will be carried out by our social workers and therapists in the SCI units to make sure that the clients are medically fit for transfer to our facility, and they understand the management plan in our facility.

There is a total of 18 places for the service. The fee for this service is HKD 1871 (about USD 240) per month after subsidy from the Government. Daily physiotherapy and occupational therapy are provided. Nurses and healthcare workers are available round-the-clock to provide basic nursing care. A case manager, who is a registered professional social worker, is assigned to each client to coordinate all the different services and to ensure that their biopsychosocial needs are met. Before the inauguration of the service, professional staff of JCNPI received training and conducted observations in the SCI units of public hospitals to equip themselves with essential knowledge and skills required for taking care of SCI patients. Consultant clinicians with expertise of management of SCI, including orthopaedic surgeons, intensivists, rehabilitation specialists, and urologists, were appointed honorary medical advisors to provide regular training updates for the staff of JCNPI. Topics discussed in these training sessions include prevention of pressure injury, autonomic dysreflexia, respiratory impairment, and neurogenic bowel and bladder dysfunctions.

JCNPI adopts an empowering and participatory approach centring on the different needs of the individual client. Unlike the experience of staying in the hospital as a patient, clients have more autonomy in their daily routines and in decision-making on the plan for their community reintegration and adaptive lifestyle. This aligns with the strategic direction of empowering and engaging people with disabilities proposed by the World Health Organization (WHO) [3]. With the regular rehabilitation therapy sessions provided by JCNPI, the clients usually improve further functionally and gain more confidence in taking back control of their own lives after SCI. In the evenings and during the weekends, clients are allowed to have leisure activities of their own choices as if they are living in the community. With this increased engagement in their daily life, the clients also understand more about the activity limitations and participation restrictions imposed by the injury on their lives. The staff will discuss with the client to work out what is most suitable to overcome these difficulties. 

With the continuation of training, the client usually experiences progressive improvement in functions and confidence in daily life. Doctors in charge of the SCI units where the client comes from will be updated regularly on their progress. If outpatient follow-up is required, JCNPI can provide accessible transportation to public hospitals with our two adapted minibuses with wheelchair lifts (Figure 2). At the same time, the team at JCNPI will explore different options of community reintegration including choices of residence and caregiver arrangement. For those clients who do not have a family available, a foreign domestic helper, for example, from the Philippines or Indonesia, is usually employed to take care of the client. Once the caregiver is identified and available, training will be initiated in JCNPI. The caregiver will be encouraged to participate in daily care and therapy sessions with the client. The case manager will assist in relocation of residence and application for welfare for the client if necessary. The physiotherapists and occupational therapists in JCNPI will also give professional inputs for prescription and training in using appropriate assistive devices and technology, as well as optimization of physical performance and modification of home environment. When the home modifications are completed and the assistive devices are delivered, the client will be ready for discharge from JCNPI and return to community living.

Taking the first step to change from an institutional care situation to home living will be daunting for our clients, despite all the preparatory work which has been done. To allow our client to have a smooth return to community living, we can arrange a two to three days’ trial stay in the pre-discharge suite in JCNPI. Equipment similar to the client’s modified residences will be provided in the pre-discharge suite to simulate the new environment which our clients will be living in (Figure 3). The caregiver will also have the opportunity to apply what they have learned from us and practice taking care of the client independently. If any problem arises during the trial stay, the client can report to the JCNPI team, and appropriate solutions will be sought with the joint efforts of professionals, the client, and the caregiver. The process can be repeated with a gradual lengthening of the stay until the client and the caregiver are physically and psychologically ready. When the discharge plan is finalized, we will inform the referring SCI unit to update the latest condition of our clients and coordinate the follow-up appointments.

### 4.3. Post-Discharge Community Day Rehabilitation Programme

Our services and relationship with the clients do not end when they are discharged from the JCNPI. We provide a post-discharge community day rehabilitation programme delivered by our physiotherapists and occupational therapists for up to 12 months for our clients. There is a total of 20 places on this programme. The charge per session is HKD 61 (about USD 8) per session. Accessible transportation between clients’ residences and JCNPI by our two adapted minibuses is available when necessary. The aim of the programme is to reinforce the clients’ competency in adapting to the new lifestyle, and to build a habit of regular exercise which will be important to maintain their health and prevent complications from SCI. If the client experiences any problems after discharge, they can also approach our team to discuss and find solutions. Refresher training is provided to caregivers when necessary, for example, when there are changes in the medical conditions or physical capacity of our clients, or when new assistive equipment needs to be acquired. Our therapists will provide updated reports of their progress to the clinicians in charge before they attend the follow-up at the SCI units. Likewise, if there are some areas which need more emphasis, the clinicians can contact our therapists to modify their training. An effective two-way cross-sector communication is maintained throughout the residential and day rehabilitation periods.

Attending the programme is not only beneficial for the physical health of our clients, but it also allows our discharged clients, or alumni as we usually call them, to come together and exchange their knowledge and experience of coping with SCI. In the past two decades, the Government has funded different NGOs to operate a range of district-based community support services for the severely disabled. According to the districts where our alumni live and their specific needs, the case manager of JCNPI can refer them to appropriate services ranging from social and recreational activities, vocational training, domiciliary rehabilitation and nursing care, personal care and escort, as well as family and caregiver support services. In addition, the Centre also has close collaborations with a network of self-help groups. We can refer our alumni to the ones which suit them best according to the missions and locations of these self-help groups. JCNPI has served the function of a hub connecting people in the SCI community with different levels of experience, and also a place to obtain useful resources and community services (Figure 4).

### 4.4. Residential Respite Service 

Taking care of individuals with SCI is no easy task. In a local survey conducted in Hong Kong in 2021, the average time spent taking care of a tetraplegic person is 11.3 h per day [4]. For those who are cared for by family members, the caregivers very often have no time to deal with their own issues. The fatigue and stress of being a caregiver should not be overlooked as it can precipitate psychological problems like anxiety and depression. On the other hand, for those who are cared for by foreign domestic helpers, the helpers may be unavailable temporarily due to annual leave, or expiry of contracts. JCNPI has four beds for residential respite service. Clients with tetraplegia are given priority and they can stay for a maximum of 30 days per year. The fee is HKD 81 (about USD 10) per day. With this service, the caregiver can take a break from their busy daily routine of taking care of SCI, and the client need not worry about unavailability of a caregiver. If the client is hiring a new domestic helper to replace the previous one, JCNPI can also provide training for the new helper.

### 4.5. The Application of the International Classification of Functioning, Disability, and Health (ICF)

The government has recognized the effective cross-sector collaboration between the medical and social welfare sectors provided by JCNPI in the first five years of its service. Funding for JCNPI has become a regular part of the social welfare budget since 2013 after the completion of the pioneer project. In the past two decades, the government has continued to strengthen the community support service for the disabled to prevent unnecessary institutional care. The Rehabilitation Advisory Committee, which is responsible for advising the government on rehabilitation policy, submitted the latest Rehabilitation Programme Plan in 2020 [5]. One of its strategic recommendations to community service providers was to “apply the ICF framework in devising a structure comprising rehabilitation objectives and intervention”.

ICF (Figure 5) is the WHO framework for measuring health and disability at both individual and population levels. It was officially endorsed by all 191 WHO Member States at the 54th World Health Assembly in 2001 as the international standard to describe and measure health and disability [6]. ICF core sets listing the essential categories relevant to SCI have been developed in the early post-acute phase [7] and the long-term context [8]. The ICF core sets for SCI were reported to have covered the majority of problems related to individuals with SCI managed by physiotherapists [9] and occupational therapists [10]. Since 2020, the multi-disciplinary team of JCNPI adopted the ICF core sets for SCI in the long-term context for the documentation, assessment, and monitoring of different kinds of impairments, activity limitations, participation restrictions, and environmental barriers. The standardized classification and documentation facilitate the communication, goal-setting and intervention planning processes among different professionals in the team [11]. The team members can have a clear understanding of the disability and rehabilitation progress, not only in those areas relevant to their own professions, but also with a holistic view of the overall function and performance of our clients.

### 4.6. Evaluation of Service Outputs and Client Outcomes

JCNPI will celebrate its 16th anniversary in 2024. We have evaluated our service outcomes by the number of clients served and the rate of successful community reintegration. Comments and feedback are collected from our service users to evaluate our service qualitatively.

### 4.7. Service Statistics

Since our opening in 2008, the mean annual number of applications for the transitional residential rehabilitation service is 19.4 (14–27). Of these, around 79% of the applications were accepted (Figure 6). About 16% of the applicants withdrew their applications prior to the completion of the intake process. The mean annual number of admissions of new cases is 14.3 (9–21). The most common reasons for withdrawal as reported by the applicants are continuous hospitalization due to medical or personal issues (67% of withdrawals), availability of modified residence sooner than the originally scheduled time (14%), and change in discharge plan to institutional care (16%). The average waiting period for admission to JCNPI in the past 16 years is around 9 weeks.

In the past 16 years, we have served a total of 243 clients. The mean occupancy is 93%. The mean number of annual re-hospitalization episodes is 12.1 (4–20). The mean annual number of discharges is 14.2 (9–17). Among the 226 clients discharged from the transitional residential rehabilitation service, 39.2% were discharged within 1 year, and 73.9% have done so within 1.5 years. The distribution of the length of stay at JCNPI is shown in Figure 7. The main reason for the delay in returning home was unsuitability of housing units for modification or the poor accessibility of the surrounding environment. Sometimes, the assigned housing units are located in older public housing estates which have less than optimal accessibility to mass public transit. When an inappropriate public housing unit is allocated to one of our clients, it may usually take at least three to four months to obtain another offer of housing from the government. The transitional residential rehabilitation service has been a success in promoting community reintegration of individuals with SCI. A total of 223 (98.6%) of the discharged clients returned to community living. Only three (1.4%) ended up in long-term institutional care due to ageing and significant family issues (Figure 8).

The post-discharge community day rehabilitation programme is also very popular among our clients. Approximately 50 to 70 discharged clients use this service annually. On average, these clients attend two half-day training sessions per week. The residential respite care serves approximately 50 clients each year for periods ranging from an overnight stay to one month.

### 4.8. Feedback from Clients

Clients who have successfully returned to community living rated their experience in JCNPI highly. They appreciated the service assisting them to improve physical capacity, regain confidence, reduce caregiver stress, and enhance participation in the community. Several of them even became our ambassadors to share their journey with peer patients in hospitals and students in schools. Here are some quotes extracted from media interviews with our clients, and appreciation letters or cards sent to us by our clients when they were discharged:-“I can now move my upper limbs after 6 months of training here. My progress is a miracle even in the eyes of my doctor.” From an elderly client with cervical spinal cord injury.-“My physical capacity improves a lot here and my confidence also increased.” From a man with cervical spinal cord injury.-“The service reduces our stress through modifying our home environment.” From parents of an adolescent with cervical spinal cord injury.-“I stand tall again soon for not wanting my parents to worry about me.” “By visiting hospitals and schools, I wish to use my life journey to uplift other peers that one can live a brilliant life despite having an injury, even though I cannot walk again.” “We will see the sunshine after the downpour.” From a man who has become our ambassador after his cervical spinal cord injury.

## 5. Discussion

### 5.1. Service Model

The service model of JCNPI is somewhat similar to the transitional rehabilitation service in Queensland, Australia. In the Australian counterpart, a multidisciplinary team including physiotherapy, occupational therapy, nursing and social work provides care to individuals who recently suffered SCI [12,13]. We both have time-limited community-based programmes and case coordinators/managers to assist our clients with adapting to living in the community. Families and caregivers can have more opportunities to participate in the final stage of preparation in returning to the community. The families or caregivers can reside in the simulation room to practice and rehearse what it would be like after discharge, which is impossible in conventional hospital rehabilitation settings. This is not only important for the family or caregiver to familiarize themselves with the daily care routine, but also helps to establish or restore their relationships, which is essential for the continuity and sustainability of care.

The main differences of the Australian service from ours include provision of service to individuals with SCI living in the metropolitan areas in their home environment and their shorter period of service (8 to 12 weeks vs. 1 year). In Hong Kong, due to the very tight living space, it is difficult to provide training to individuals with SCI at their homes in the early period of rehabilitation. Clients usually need longer periods of time to get their home modifications completed and assistive equipment delivered. In contrast to the geography of Queensland, most of the people in Hong Kong can access services of major regional hospitals within 60 minutes’ travel. Accidents and Emergency Departments can often be reached within 15 to 30 minutes’ travel. The public healthcare system provides 90% of medical services in Hong Kong, therefore it is easier for us to establish a close and well-coordinated working relationship. We do not have the problem of educating and liaising with local care providers.

### 5.2. Facilitators of and Barriers to SCI Transitional Care

JCNPI manages to keep the utilization rates of all services at above 98% after operation for 16 years. Cross-sector collaboration between social sector (JCNPI and its parent organization SAHK) and medical sector (SCI units in public hospitals) has been the key to its success. Establishment of an effective two-way communication and knowledge exchange channel has helped grow the Centre’s confidence of dealing with the most complicated biopsychosocial problems arising from SCI. The enormous financial support from charity (the Hong Kong Jockey Club Charities Trust) for the initial funding, and the government for the ongoing operational budget, are also indispensable.

There are several barriers identified. The one which has the most significant impact on our service delivery is manpower shortage. On the one hand, NGOs and public hospitals have collaborated extensively to serve the citizens of Hong Kong. On the other, they are also competitors in the market of employment of healthcare assistants and therapists. Due to the limitations of budget and service scope, NGOs cannot provide clinical exposure, salary and benefit packages, and career paths comparable to those offered by public hospitals, making positions in NGOs less attractive and resulting in a high turnover rate among our healthcare professionals. In the past decade, the government, through the Social Welfare Department, has funded Master of Physiotherapy and Master of Occupational Therapy programmes to provide entry-level training for holders of undergraduate degrees of other disciplines to pursue careers in rehabilitation professions. The graduates from these programmes must spend their first three years of service in an NGO. The initiative has alleviated the manpower shortage of rehabilitation professionals, which is especially severe in NGOs.

Another factor affecting the efficient running of our community day rehabilitation programme is inadequate accessible transport services. In Hong Kong, owning a car is considered a luxury due to the high cost of fuel, maintenance and parking costs, despite the government subsidy for vehicles used by the disabled. Though the mass transit system in Hong Kong has accessible features, the client may have difficulty in travelling from their homes to the mass transit station. JCNPI can provide limited point-to-point accessible transport with our two adapted minibuses, but this may not fulfil all the needs of our clients. Some clients occasionally have to skip their scheduled sessions due to lack of transportation. Despite the efforts of SCI units in referring suitable cases to JCNPI, some individuals with SCI or their families are not willing or not ready to consider community living. Some may want to receive more in-patient rehabilitation in the hospital, some may be seeking alternative or traditional Chinese medicine treatment, and some have not yet accepted the disabilities caused by their injuries. The daily hospital fee for Hong Kong residents, including all medications, nursing care, and procedures, is HKD 60 (about USD 8), and there is little financial incentive for them to leave the hospital.

### 5.3. Challenges and Opportunities Ahead

The economy has not rebounded as expected after the prolonged lockdown of the city during the COVID-19 pandemic. Most NGOs in Hong Kong are expecting cuts in government funding in the coming years. Administrators are pessimistic and very cautious in planning budgets. Moreover, many experienced clinicians and therapists working in public hospitals are approaching their retirement. The three SCI units will face problems in finding suitable successors. The increased workload in public hospitals also limits their support for training and exchange of knowledge with us. However, the significant savings to the public hospital system contributed by JCNPI have been proven. With a mean annual subvention of HKD 7,798,000 (about USD 1 million) from the government and a mean occupancy of 93% in the past 16 years, the annual cost per client of the transitional residential service is calculated to be HKD 466,000 (about USD 60,000). The daily public hospital charge for non-Hong Kong residents is set at HKD 5100 (about USD 650) to recover costs. The annual cost of an in-patient is HKD 1,861,500 (about USD 239,000). The annual saving of public resource per client is HKD 1,395,500 (about USD 178,900). There is, therefore, clear financial benefit, in addition to the empowerment and improved independence of individuals with SCI, in supporting the continuing use of the transitional care service rather than public hospital beds. Due to the current limitations of the service, as well as of this paper, we have not evaluated the transitional care of SCI in Hong Kong with standardized outcome measures on quality of life like the SF-36. Further quantitative analysis of the service in the future will be helpful for stakeholders to appreciate its effectiveness.

While we are facing challenges, we are also seeing new opportunities. People became used to teleconsultation, online training and video conferencing during the pandemic. The use of such technology may enhance service and connection with our clients when our physical capacity cannot be increased or when transportation is not available. Applications on mobile devices for transitional care of home-dwelling SCI patients have been developed in mainland China to support them after discharge [14,15]. Time and travel costs can also be saved in staff training and experience sharing. Collaboration with professionals overseas has also become easier.

## 6. Conclusions

JCNPI has transformed the care for individuals with SCI in Hong Kong from the previous biomedical-oriented, hospital-based rehabilitation into a journey with an empowering and participatory approach addressing their biopsychosocial needs. The model has proven to be a key player in the continuum of care and sustainable community reintegration of individuals with SCI.

## Figures and Tables

**Figure 1 healthcare-12-02388-f001:**
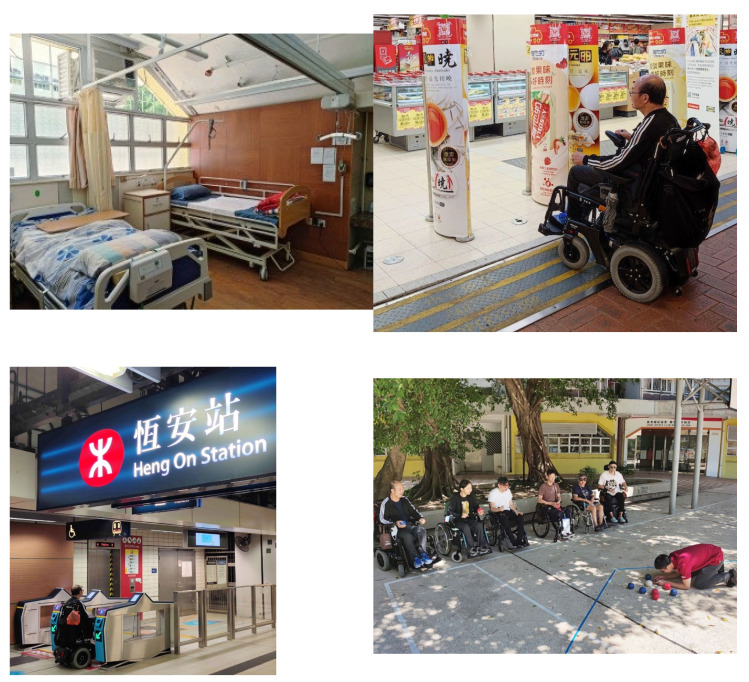
The Jockey Club New Page Inn (JCNPI) and the community facilities in the surroundings. Left upper: typical setup of a bedroom for transitional residential service; Right upper: a supermarket in the neighbourhood; Left lower: the nearby Mass Transit Railway station; Right lower: clients enjoying boccia under guidance by the staff at the basketball in front of JCNPI.

**Figure 2 healthcare-12-02388-f002:**
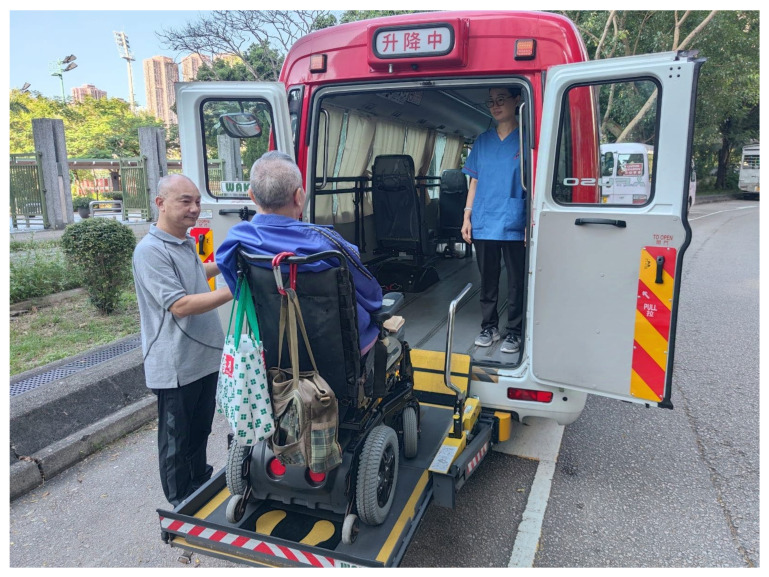
The JCNPI has two modified minibuses equipped with wheelchair lifts. The Chinese characters at the top of the vehicle means “wheelchair lift in operation”.

**Figure 3 healthcare-12-02388-f003:**
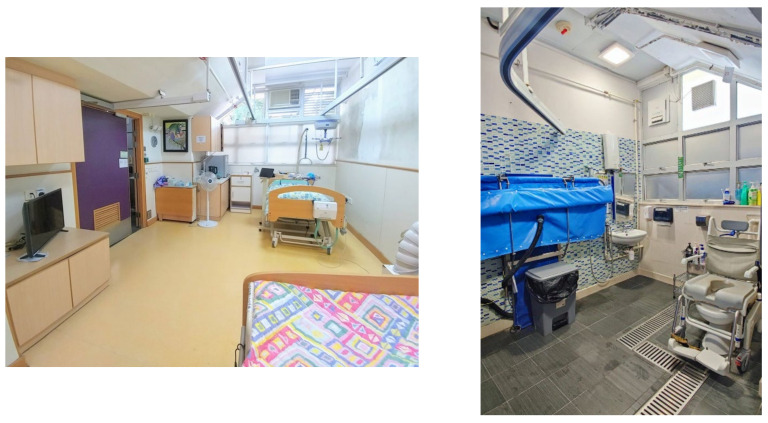
The pre-discharge suite for short period trial stays by the client and caregiver.

**Figure 4 healthcare-12-02388-f004:**
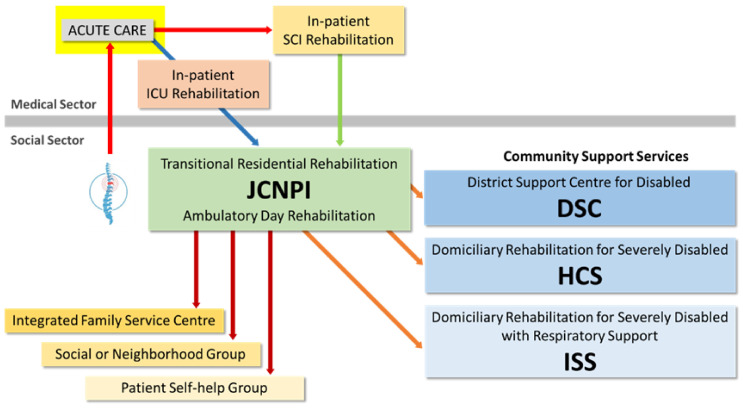
JCNPI serves as a hub connecting the public hospitals with community support, family and caregiver support, and peer networking services for survivors with SCI.

**Figure 5 healthcare-12-02388-f005:**
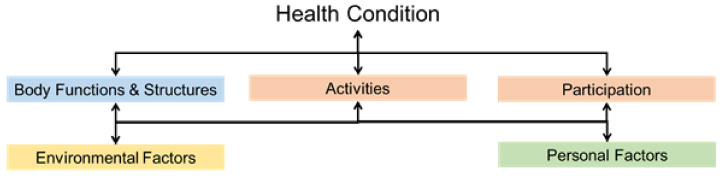
The five-component biopsychosocial model of functioning, disability and health. (Adapted from International classification of functioning, disability and health: ICF. Geneva: World Health Organization. 2001. CC BY-NC-SA 3.0 IGO [6]).

**Figure 6 healthcare-12-02388-f006:**
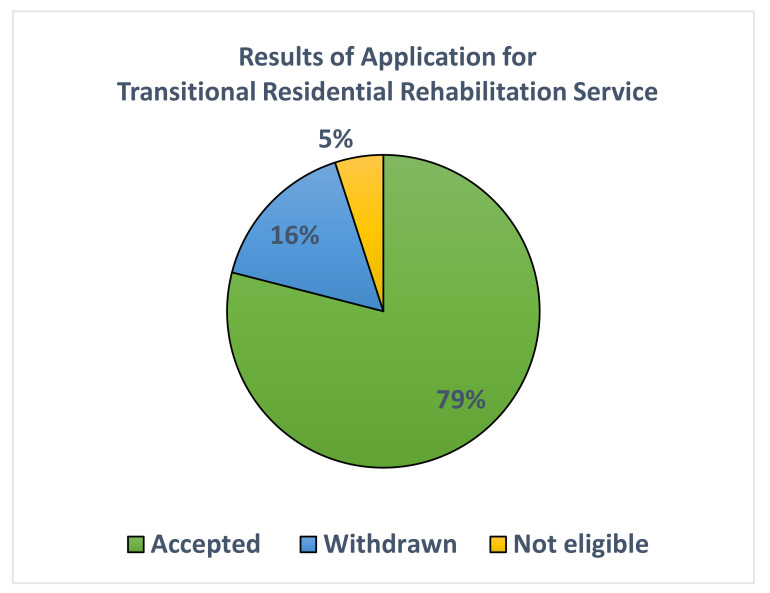
Results of application for transitional residential care.

**Figure 7 healthcare-12-02388-f007:**
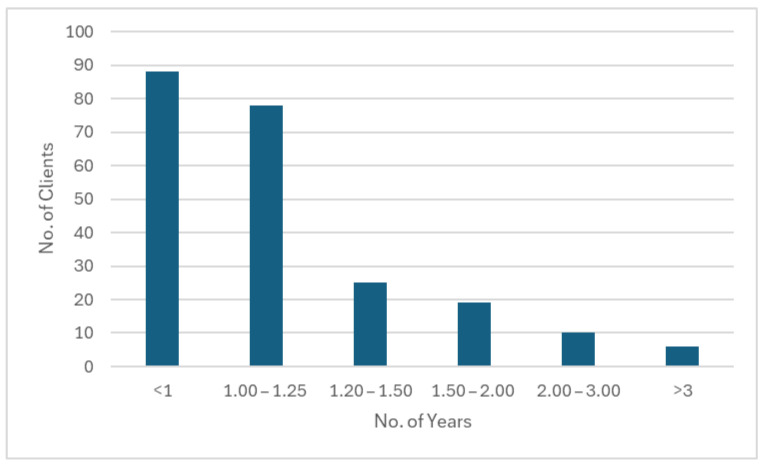
Distribution of length of stay (in years) of the residents of JCNPI in the past 16 years.

**Figure 8 healthcare-12-02388-f008:**
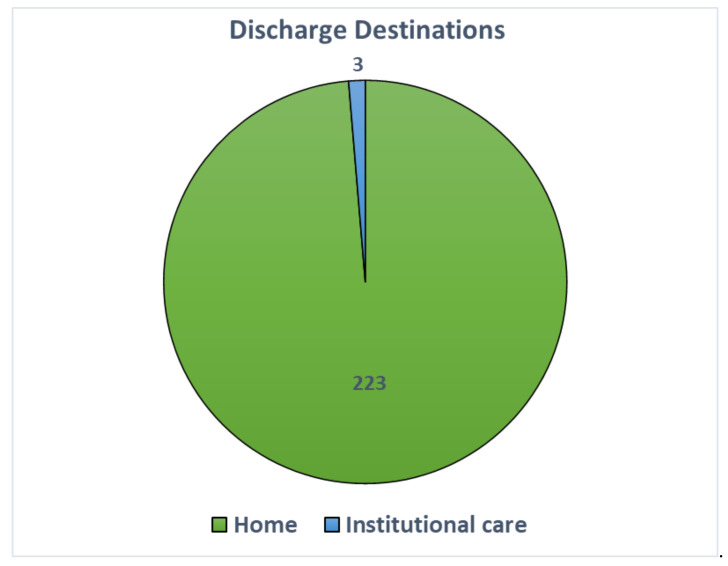
Discharge destinations of transitional residential rehabilitation clients.

## Data Availability

The original contributions presented in the study are included in the article, further inquiries can be directed to the corresponding author.

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
