# Peer review of "Transitional Care for Spinal Cord Injuries in Hong Kong SAR, China: A Narrative Review of the Local Experience"

_healthcare, 2024, doi:10.3390/healthcare12232388_

Round 1
Reviewer 1 Report
Comments and Suggestions for Authors
Dear Author,
The authors aimed to review the local experience on the implementation of SCI transitional care in Hong Kong SAR, China. I would like to congratulate the authors for writing this article. I think that the article will shed light on academicians, clinicians working in this field and patients.
My only suggestion to the authors is that they used too many abbreviations, which causes some difficulties in reading. if possible, I suggest that they write the exact terms instead of using abbreviations in some places. I couldn't see a clear explanation of the abbreviation “SHAK” in the text.
Additional comments:
I think the article was written very successfully, the introduction, method parts of the article were written appropriately.
References were selected appropriately for the article. The language and the expression are appropriate for the readers to understand.
Author Response
Dear Reviewer 1,
Comment 1: My only suggestion to the authors is that they used too many abbreviations, which causes some difficulties in reading. if possible, I suggest that they write the exact terms instead of using abbreviations in some places.
Response 1: Thank you for pointing it out. We have checked the manuscripts. The abbreviations we have used include SCI (Spinal Cord Injury), NGO (Non-Governmental Organization), JCNPI (Jockey Club New Page Inn), WHO (World Health Organization), ICF (International Classification of Functioning, Disability and Health).
We think most of these terms (except JCNPI) are all commonly used and acceptable to the readers.
We think that it will be easier for the readers to read our article if we use the abbreviation JCNPI because it has to appear many times during our writing.
Comment 2: I couldn't see a clear explanation of the abbreviation “SHAK” in the text.
Response 2: Thank you for pointing it out. The background and the development of the non-governmental organization from “The Hong Kong Association for the Spastic Children” to “SAHK” is described in line 59-64. SAHK is currently the official name of the organization. The name has been chosen because it reflects the expansion of its service beyond the paediatric group, avoids stigmatization of the clients and, at the same time, retains its identity and long history of service in Hong Kong.
Explanation of using the name “SAHK” has been added to the manuscript Line 64-67.
Reviewer 2 Report
Comments and Suggestions for Authors
The manuscript entitled “Transitional Care for Spinal Cord Injuries in Hong Kong SAR, China: A Narrative Review of the Local Experience” is an original article. It covers a wide range of JCNPI services in Hong Kong SAR, China. It addresses social care in the rehabilitation of SCI patients, particularly in transitional care. The article offers a comprehensive overview and presents valuable information about transitional care for SCI patients; however, it could be enhanced in several areas.
Areas for Improvement:
1. Background
a. The incidence of SCI patients has been presented without gender and age specificity. Including this information may provide a deeper understanding of the studied parameters.
2. Results
a. Do foreign domestic helpers receive training to learn the local language for effective communication and to prevent miscommunication?
b. The feedback and survey formats, along with the application format, can be added as supplementary material for further reference, adding more authenticity.
c. Yearly service statistics should be included to track the number of applications accepted, rejected, or withdrawn, and to determine if there is any significant difference in reasons for withdrawn applications across different years.
3. Discussion
a. The findings of the transitional care and satisfaction survey should be discussed further to improve this section’s efficiency and enhance its readability.
The manuscript is well-crafted and of high quality. Applying the suggested comments will greatly enhance it.
Author Response
Dear Reviewer 2,
Comment 1: The incidence of SCI patients has been presented without gender and age specificity. Including this information may provide a deeper understanding of the studied parameters.
Response 1: Thank you for your comments. Since this is not a paper focusing on the epidemiology of SCI in Hong Kong, we did not include more details in our manuscript. However, we have added more information on the gender and age in line 37-38.
Comment 2: Do foreign domestic helpers receive training to learn the local language for effective communication and to prevent miscommunication?
Response 2: Thank you for the question.
Foreign domestic helpers received training on local language and basic training of their work by their employment agents before coming to Hong Kong. We will provide further training on taking care of SCI to them in our centre as mentioned in line 150-158.
Comment 3: The feedback and survey formats, along with the application format, can be added as supplementary material for further reference, adding more authenticity.
Response 3: Thank you for your suggestions. The clients’ comments are mainly from media interviews of our clients, and the appreciation letters or cards we received. They are not from a formal survey. Information has been added to line 304-306.
Comment 4: Yearly service statistics should be included to track the number of applications accepted, rejected, or withdrawn, and to determine if there is any significant difference in reasons for withdrawn applications across different years.
Response 4: Thank you for your suggestion. We have added information on the number of applications (line 263-267, line 276-278).
Comment 5: The findings of the transitional care and satisfaction survey should be discussed further to improve this section’s efficiency and enhance its readability.
Response 5: Thank you for your comments. Information has been added to line 304 – 306 to improve clarity.
Reviewer 3 Report
Comments and Suggestions for Authors
This article provides a comprehensive overview of the transitional care model for spinal cord injury implemented in Hong Kong SAR through the Jockey Club New Page Inn. It highlights the services offered, including residential rehabilitation, community day programs, and respite care, and evaluates the outcomes, notably the high rate of community reintegration (98.6%). The authors discuss the collaborative efforts between public healthcare and NGOs that have contributed to the success of this model. However, there are areas where additional data and comparisons could enhance the article’s robustness and facilitate better international benchmarking.
Overall, the article is well-structured and presents an impressive case for the effectiveness of the JCNPI in SCI transitional care.
Recommendations for Improvement:
-
To strengthen the analysis and enhance the comparability of the program’s cost-effectiveness, I recommend the inclusion of cost data adjusted for PPP. This would allow readers to contextualize the financial commitment and make cross-border evaluations more feasible.
-
It would be beneficial to include the annual average occupancy rate of the JCNPI. This metric would provide insights into the utilization of resources and indicate the program’s operational efficiency.
-
Introducing standardized outcome measures such as the SF-36 would enrich the discussion on patient-reported quality of life post-rehabilitation. Additionally, data on the hospital readmission rate post-discharge could serve as a critical indicator of the program’s long-term efficacy and stability in community reintegration.
Author Response
Dear Reviewer 3,
Comment 1: To strengthen the analysis and enhance the comparability of the program’s cost-effectiveness, I recommend the inclusion of cost data adjusted for PPP. This would allow readers to contextualize the financial commitment and make cross-border evaluations more feasible.
Response 1: Thank you for your comments.
The cost data is added to the section of discussion line 390-398.
Comment 2: It would be beneficial to include the annual average occupancy rate of the JCNPI. This metric would provide insights into the utilization of resources and indicate the program’s operational efficiency.
Response 2: Thank you for your comments. The information is added to service statistics section (line 263 – 278).
Comment 3: Introducing standardized outcome measures such as the SF-36 would enrich the discussion on patient-reported quality of life post-rehabilitation. Additionally, data on the hospital readmission rate post-discharge could serve as a critical indicator of the program’s long-term efficacy and stability in community reintegration.
Response 3: We agree with your comments. Since the aim of our paper is to explore the model of service, the facilitators and barriers, and potential future development, we do not have the QoL measures this time. We agree that it will be a very useful study to include these measures in the future.
We have the statistics of readmission to hospital during their stay in our facility and it is added to the service statistics section (line 276-278). We do not have the post-discharge readmission statistics because it is only available from the hospital. We agree that it will be useful to assess the long-term sustainability of community reintegration. We may collaborate with the public hospitals to study these data in the future.
Reviewer 4 Report
Comments and Suggestions for Authors
Thank you for the opportunity to review a paper titled Transitional Care for Spinal Cord Injuries in Hong Kong SAR, 2 China: A Narrative Review of the Local Experience
I read the article with attention and interest. The authors describe the services offered and the facilities in detail. Unfortunately, this paper does not meet scientific standards. It is closer to a marketing presentation than a scientific analysis.
The paper should be classified as a review. The authors claim in the methods section: ”We performed a narrative review of the service model, facilitators and barriers, and 69 future development of transitional care in Hong Kong SAR, China”. Line 69-70
Client feedback should not be included in scientific papers. You can measure QofL using scientific methodology, not marketing methods.
Please consider changing the structure and aim of the paper. You have patients and statistics. In the introduction, present your services and facilities. Still, the paper should refer to the patient group. In the material section, you may describe some characteristics—nr, sex, level of injury, cause of injury, ASIA scale, QofL, etc., methodology—tools, measurements, and results— for example, how rehab influences the outcomes. Statistics referred only to the application is a marketing field of interest, not medical.
Author Response
Dear Reviewer 4,
Comment 1: Unfortunately, this paper does not meet scientific standards. It is closer to a marketing presentation than a scientific analysis.
The paper should be classified as a review. The authors claim in the methods section: ”We performed a narrative review of the service model, facilitators and barriers, and 69 future development of transitional care in Hong Kong SAR, China”. Line 69-70
Client feedback should not be included in scientific papers. You can measure QofL using scientific methodology, not marketing methods.
Response 1: Thank you for your comments. We agree that we are not writing a scientific paper to measure quantitatively the work of our SCI translational service. Instead, we have positioned ourselves in writing a review of our service model, facilitators and barriers, and possible future development locally as stated in our aims.
Comment 2: Please consider changing the structure and aim of the paper. You have patients and statistics. In the introduction, present your services and facilities. Still, the paper should refer to the patient group. In the material section, you may describe some characteristics—nr, sex, level of injury, cause of injury, ASIA scale, QofL, etc., methodology—tools, measurements, and results— for example, how rehab influences the outcomes. Statistics referred only to the application is a marketing field of interest, not medical.
Response 2: Thank you for your comments. We agree that all the parameters you have mentioned will be important for further evaluation of our service. We plan to explore further in these aspects in the future.
Round 2
Reviewer 4 Report
Comments and Suggestions for Authors
My comments were not taken into account, there are still phrases not used in scientific articles
All 3 authors have been significantly involved in advisory, supervisory and executive 78
roles in the local SCI transitional care. CYL is a honorary medical advisor of SAHK. IWYS 79
is the head of professional and programme development of SAHK. JYML is a senior man- 80
ager of SAHK and former superintendent of JCNPI.
How does it relate to methodology?
In the line 303-317; Client's opinions are NOT scientifical methods. Changes made by authors are not major. In this form it does not look like scientific paper for the journal with IF.
Best regards
Author Response
Dear Reviewer 4,
Comment 1: All 3 authors have been significantly involved in advisory, supervisory and executive 78
roles in the local SCI transitional care. CYL is a honorary medical advisor of SAHK. IWYS 79
is the head of professional and programme development of SAHK. JYML is a senior man- 80
ager of SAHK and former superintendent of JCNPI.
How does it relate to methodology?
Response 1: Thank you for your comments. In the methodology part, we mentioned that the “Information on the transitional care service was obtained with approval from the service provider (SAHK).” (Line 76-77). We included this information so that the readers would understand the involvement and perspectives of the authors.
Comment 2: In the line 303-317; Client's opinions are NOT scientifical methods. Changes made by authors are not major. In this form it does not look like scientific paper for the journal with IF.
Response 2: We do not agree completely with the comment. Though the origin of including quotations in qualitative research remains uncertain, and limited guidance in the scientific literature concerning quotations is noted. Various authors have supported the use of quotations in academic papers, especially those involving social science subjects and public health. By including the clients own words in the article, we think that it adds the personal experiences of the changes we have made on our clients.
However, if it is the editorial policy of the journal not to include qualitative evidence/ quotations from users, we are willing to remove the part from our article.
References:
Eldh, A. C., Årestedt, L., & Berterö, C. (2020). Quotations in Qualitative Studies: Reflections on Constituents, Custom, and Purpose. International Journal of Qualitative Methods, 19.
White C., Woodfield K., Ritchie J., Ormston R. (2014). Writing up qualitative research. In Ritchie J., Lewis J., Nicholls McNaughton C., Ormston R. (Eds.), Qualitative research practice (pp. 368–400). Sage.
Corden A., Sainsbury R. (2006). Exploring “quality”: Research participants’ perspectives on verbatim quotations. International Journal of Social Research Methodology, 9(2), 97–110.